# Longitudinal Analysis and Comparison of Six Serological Assays up to Eight Months Post-COVID-19 Diagnosis

**DOI:** 10.3390/jcm10091815

**Published:** 2021-04-21

**Authors:** Aurélien Aubry, Baptiste Demey, Catherine François, Gilles Duverlie, Sandrine Castelain, François Helle, Etienne Brochot

**Affiliations:** 1Department of Virology, Amiens University Medical Center, 80000 Amiens, France; aubry.aurelien@chu-amiens.fr (A.A.); demey.baptiste@chu-amiens.fr (B.D.); catherine.francois@u-picardie.fr (C.F.); gilles.duverlie@u-picardie.fr (G.D.); sandrine.castelain@u-picardie.fr (S.C.); francois.helle@u-picardie.fr (F.H.); 2Resistant Infectious Agents and Chemotherapy Research Unit, AGIR UR4294, Jules Verne University of Picardie, 80000 Amiens, France

**Keywords:** SARS-CoV-2, COVID-19, serological assays, assay performance

## Abstract

Background: There is much data available concerning the initiation of the immune response after SARS-CoV-2 infection, but long-term data are scarce. Methods: We thus longitudinally evaluated and compared the total and neutralizing immune response of 61 patients to SARS-CoV-2 infection up to eight months after diagnosis by RT–PCR using several commercial assays. Results: Among the 208 samples tested, the percentage of seropositivity was comparable between assays up to four months after diagnosis and then tended to be more heterogeneous between assays (*p* < 0.05). The percentage of patients with a neutralizing titer decreased from 82% before two months postdiagnosis to 57% after six months. This decrease appeared to be more marked for patients under 65 years old and those not requiring hospitalization. The percentage of serology reversion at 6 months was from 11% with the WANTAI total assay to over 39% with the ABBOTT IgG assay. The neutralizing antibody titers decreased in parallel with the decrease of total antibody titers, with important heterogeneity between assays. Conclusions: In conclusion, serological tests show equivalent sensitivity in the first months after the diagnosis of SARS-CoV-2 infection, but their performance later, postinfection, must be considered when interpreting the results.

## 1. Introduction

COVID-19, defined as the disease caused by the new SARS-CoV-2 coronavirus, has now caused more than two million deaths worldwide, and more than 100 million cases have been detected by RT-PCR. The SARS-CoV-2 RNA encodes four structural proteins: spike (S), membrane (M), envelope (E), and nucleocapsid (N). After infection by the virus, an immune response directed against these proteins is rapidly established during the first two weeks following infection, making it possible, in most cases, to effectively control the infection. The detection of this immune response by in-house or commercial serological assays is indicative of a previous infection by SARS-CoV-2. Various groups throughout the world, including ours, have characterized the kinetics in the immune response induction against different viral proteins, with disparities depending on the severity of the infection, sex, and age of the patients [1,2,3,4]. Moreover, the detection of such antibodies for forms with few symptoms can depend on the type of antibodies sought, the antigens used for their detection, and the intrinsic performance of the assay [5,6]. Firms have adopted various strategies in terms of the choice of antigenic base and the type of immunoglobulins detected. Moreover, the fraction of neutralizing antibodies directed against the S protein found in patients does not always correlate with the concentrations or index found using such commercial assays [7]. Decreases in antibody titers distant from the infection will inexorably occur, with kinetics that are not currently known. It is, therefore, important to know the rate of false-negative serological results several months after infection with SARS-CoV-2 for each assay, as well as the kinetics of the decrease in neutralizing antibody titers. Here, we evaluate the kinetics of antibody titers among 61 patients (208 samples) up to eight months after SARS-CoV-2 infection in a head-to-head comparison of six commercial assays widely used worldwide, and we determine the neutralizing antibody (Nab) titers in parallel.

## 2. Materials and Methods

### 2.1. Study Design and Cohort

The study was conducted at the Amiens University Medical Center following approval by its institutional review board (number PI2020_843_0046, 21 April 2020). Samples (n = 209) were derived from deidentified excess serum specimens sent to our clinical virology laboratory. The assays were validated using serum samples from hospitalized or nonhospitalized patients (n = 61) with PCR-confirmed SARS-CoV-2 infections.

### 2.2. Serological Assays

The characteristics of the various serological tests evaluated are presented in Appendix A. The antigen used in the assay was the SARS-CoV-2 nucleocapsid for the ABBOTT^®^ and BIORAD^®^ assays (Marnes-la-Coquette, France), Spike 1 for the EUROIMMUN^®^ assay (Bussy-Saint-Martin, France), and the spike receptor-binding domain (RBD) for the WANTAI^®^ assay(Changping District, Beijing, China). The ABBOTT^®^, EUROIMMUN^®^, and LIAISON-DIASORIN^®^ assays (Antony, France) detect immunoglobulin G, whereas the BIORAD^®^ and WANTAI^®^ assays detect total antibodies using a double antigen bridging assay (DABA). The Innobiochips COVIDIAG^®^ assay (Lille, France) detects various anti-SARS-CoV-2 IgG antibodies against five viral antigens. Samples with a doubtful signal were tested a second time for all assays, and the result was considered negative if it was still the same. We evaluated the specificity of these serological assays on prepandemic samples in a previous study (5).

### 2.3. Neutralization Assay

Retroviral particles pseudotyped with the S glycoprotein of SARS-CoV-2 (SARS-CoV-2pp) were produced, as previously described [1], with a plasmid encoding a human codon-optimized sequence of the SARS-CoV-2 spike glycoprotein (accession number: MN908947). Supernatants containing the pseudotyped particles were harvested at 48, 72, and 96 h after transfection, pooled, and filtered through 0.45-μm pore-sized membranes. Neutralization assays were performed by preincubating SARS-CoV-2pp and serially diluted plasma for 1 h at room temperature before contact with 293T cells (ATCC^®^ CRL-3216TM) transiently transfected with the plasmids pcDNA3.1-hACE2 24 h before inoculation. Luciferase activity was measured 72 h postinfection, as indicated by the manufacturer (Promega). Two independent tests were carried out each time in duplicate. The NAb titers were defined as the highest dilution of plasma resulting in a 90% decrease in infectivity. We previously controlled the specificity of our neutralization assay using not only plasmas from patients seropositive for other coronaviruses but also retroviral particles pseudotyped with the G glycoprotein of the vesicular stomatitis virus [1].

### 2.4. Data Analysis and Statistical Analyses

The demographic information of the 61 patients was extracted from the patient data software (detailed in Table 1). Quantitative variables are expressed as means and were compared using Student *t*-tests. The chi-square test (X2 test) and Fisher’s exact test were used to determine whether there was a significant relationship between qualitative variables. The Pearson correlation coefficient was used to measure the strength of any linear association between two quantitative variables. Statistical analyses were performed using GraphPad Prism 5 (San Diego, California). A two-sided *p*-value <0.05 was considered statistically significant.

## 3. Results

### 3.1. Head-to Head-Comparison of Serological and Neutralizing Assays

In total, 61 patients, representing 208 samples diagnosed with SARS-CoV-2 infection by RT–PCR between March and April 2020, were screened for the study. Their demographic characteristics are presented in Table 1. Twelve patients in our study were fifty years old or younger. The choice of the age threshold of 65 years is based on previous publications on the subject and the data from COVID-19 vaccine studies, highlighting this threshold for data interpretation. We analyzed the 208 samples from the cohort using the 6 serological assays described in Appendix A. Neutralizing antibody titers were also evaluated. The results were categorized according to the time of sample collection after an initial positive SARS-CoV-2 RT–PCR (0–59, 60–119, 120–179, and >180 days post-PCR). We clearly observed seropositivity for 80% to 90% of the samples up to 120 days post-PCR for all six assays, followed by a subsequent decrease for the ABBOTT assay to 76% (38/50) between Days 120 to 179 and 64% (37/58) after >180 days (Figure 1A). Statistical analysis of the six serological assays showed a significant difference for the 120–179-day period (chi-square test *p*-value: 0.048) and >180 days (chi-square test *p*-value: 0.01) post-PCR. The percentage of samples positive for NAbs also showed a progressive decrease from 82% (41/50) at 0–59 days post-PCR to 76 (38/50), 64 (32/50), and 57% (33/58) at 60–119, 120–179, and >180 days post-PCR, respectively. We performed the same analysis in Figure 1B based on the percentage of positivity for the 61 patients in the cohort. We found similar profiles, demonstrating a difference between the assays for the evaluation of seropositivity towards SARS-CoV-2 distant from infection, as well as a progressive decrease in neutralizing titers (comparison of the six assays; chi-square test *p*-value 0.03 and 0.02 for the periods of 120–179 and >180 days, respectively). 

We also examined the seropositivity for the different assays according to sex, age (>65 or <65 years), and whether the patients required hospitalization or not after diagnosis of the infection (Appendix A). We did not observe any difference depending on the sex of the patient for the same serological assay, irrespective of the post-PCR period. However, for patients under 65 years old, seropositivity was lower for the six assays by an average of −4.3%, −27.5%, −12.5%, and −17% for the periods of 0–59, 60–119,120–179, and >180 days post-PCR, respectively (Appendix A). The percentage of positive patients for Nabs did not differ according to the age of the patient. Similarly, patients who did not require hospitalization following a diagnosis of SARS-CoV-2 infection had average seropositivity of −3.6%, −5.8%, −5%, and −15% for the periods of 0–59, 60–119, 120–179, and >180 days post-PCR, respectively. We also observed the same trend for the neutralization titer assay (Appendix A).

All the raw results of the different serological assays according to the post-PCR period are presented in Appendix A. We clearly observed a decrease in the mean index over time for the EUROIMMUN (A and G) and ABBOTT assays. Conversely, the mean index rose or remained stable for up to 180 days post-PCR for the assays testing for total antibodies (WANTAI and BIORAD) and then appeared to decrease slightly after 180 days. For the Innobiochips assay, which analyzes the immune response against five viral antigens, the mean value was also stable up to 180 days and then declined slightly. The mean neutralizing antibody titers gradually declined from 112 for patient samples less than 60 days after PCR diagnosis to 48 for those after more than 180 days after PCR diagnosis (Student *t*-test *p*-value: <0.001).

### 3.2. Decline of Antibody Levels Six Months Postdiagnosis

We then compared the indexes or titers obtained by the different assays on the last sample obtained before Day 60 and the last sample obtained after Day 180 post-first positive PCR (Figure 2). The choice of the latest sample before Day 60 was made to avoid having patients who were still in the phase of rising serum antibodies.

For the 43 patients with these data, there was a clear decrease of the indexes and titers, with considerable heterogeneity between assays. The results of Student’s *t*-tests for the comparison of the average index between the two periods were 0.006, 0.29, <0.001, 0.37, 0.43, 0.2, and <0.001 for the EUROIMMUN A, EUROIMMUN G, ABBOTT, WANTAI, BIORAD, INNOBIOCHIPS and neutralizing assays, respectively. We observed 16, 19, 3, 41, and 32 patients who did not show a decrease in the index between these two samples for the EUROIMMUN A, EUROIMMUN G, ABBOTT, WANTAI, and BIORAD assays, respectively. This comparison was not possible for the INNOBIOCHIPS assay, which is based on an average of different measurements. For patients with a neutralizing titer before D60, only 8/35 did not show a decrease. For the patients who were seropositive for SARS-CoV-2 before D60, we then evaluated the percentage of reversion after Day 180 according to the assay used and then sex, age, and the need for hospitalization or not for COVID-19 (Table 2). The percentage of reversion was clearly the same for men and women, regardless of the assay used. However, the percentage of reversion after 180 days was higher for nonhospitalized patients and those under 65 years old for all assays.

### 3.3. Link Between Neutralizing Titer and Serological Assays

Finally, we performed scatter plots for the 208 samples for each serological assay used to evaluate a link between the neutralizing titer and the index values (Figure 3). With the exception of the EUROIMMUN assay, which analyzes IgA, the other serological assays showed an increase in neutralizing titer as the index increased. However, we observed high variability and a wide range of index values for low neutralizing titers.

## 4. Discussion

In this study of 61 patients, followed from a positive RT–PCR for SARS-CoV-2 for up to eight months, we observed a decrease in seropositivity from six months onwards for most of the assays tested. However, the ABBOTT assay appeared to show lower performance, especially after 120 days. Other groups have already observed this phenomenon in an earlier post-PCR period [8,9]. At the end of the summer of 2020, ABBOTT defined a doubtful zone for the interpretation of its results, probably showing poor adjustment of their positivity threshold. More recently, they have developed a new serological assay for the quantification of anti-spike antibodies. This difference in terms of positivity, which can reach more than 20% between two assays over a period of time, poses a problem for comparing studies evaluating the seroprevalence of a population. The results must be interpreted with precaution, depending on the assay used and the time frame of the studies in relation to the epidemic waves for each country. Moreover, mathematical models used to extrapolate these seroprevalence data may be highly influenced by such disparities, as observed for seroprevalence studies in the city of Manaus in Brazil [10,11].

We have also observed that the mean values or index measured over time declined much more slowly for assays that test for total antibodies (IgM, IgA, and IgG) or antibodies against multiple viral epitopes. However, it must be kept in mind that these index values are not truly quantitative values and can only be compared for the follow-up of a patient using the same assay. With the deployment of SARS-CoV-2 vaccines using the spike protein as an immunogen, commercial assays for the quantification of serum antibodies against the spike protein have been available since early 2021. However, until an international standard is available, comparing results between assays may be misleading.

We also measured reversion rates six months after the diagnosis of infection (Table 2). Again, we observed reversion values from 11% (WANTAI) to 39% (ABBOTT). We did not observe any difference between men and women for the same assay but did observe a difference between individuals aged less or more than 65 years and those hospitalized or not for COVID-19. These two criteria are strongly linked, as a greater proportion of people over 65 years old are hospitalized for COVID-19 [12].

Finally, we measured neutralizing titers for the same samples using pseudoviruses and found an almost systematic decrease over time (Figure 2G). The same profile should be observed after vaccination, with, probably, a different pattern depending on the peak of neutralizing antibodies obtained and the type of vaccines administered. Similarly, we attempted to correlate the index values found with the neutralizing titer for each assay (Figure 3). We obtained a high level of disparity for the same titer but a positive correlation between these two parameters. No protective neutralizing threshold has been defined for protection against a new infection and/or a severe form, and even if a reversion for all these tests is observed, this would not automatically mean a lack of protection, taking into account immune memory and cellular immunity [13,14]. However, with the detection and emergence of multiple variants throughout the world [15], it cannot be excluded that some serological tests will underperform. As mentioned in Table 1, six immunocompromised patients were included in our study; one marrow transplant patient never seroconverted regardless of the assay, the two patients who received chemotherapy showed late seroconversion (after D60), and the other patients showed seroconversion but with a low index.

## 5. Conclusions

The current serological tests show equivalent sensitivity in the first months after the diagnosis of SARS-CoV-2, but their varying performance in later postinfection periods must be considered for interpretation of the results. Such declines in analytical reactivity and neutralizing titers after SARS-CoV-2 infection must be considered in parallel with country-specific vaccine strategies and country-specific variants. Our results suggest that using some serological assays may significantly underestimate the herd immunity of the studied populations.

## Figures and Tables

**Figure 1 jcm-10-01815-f001:**
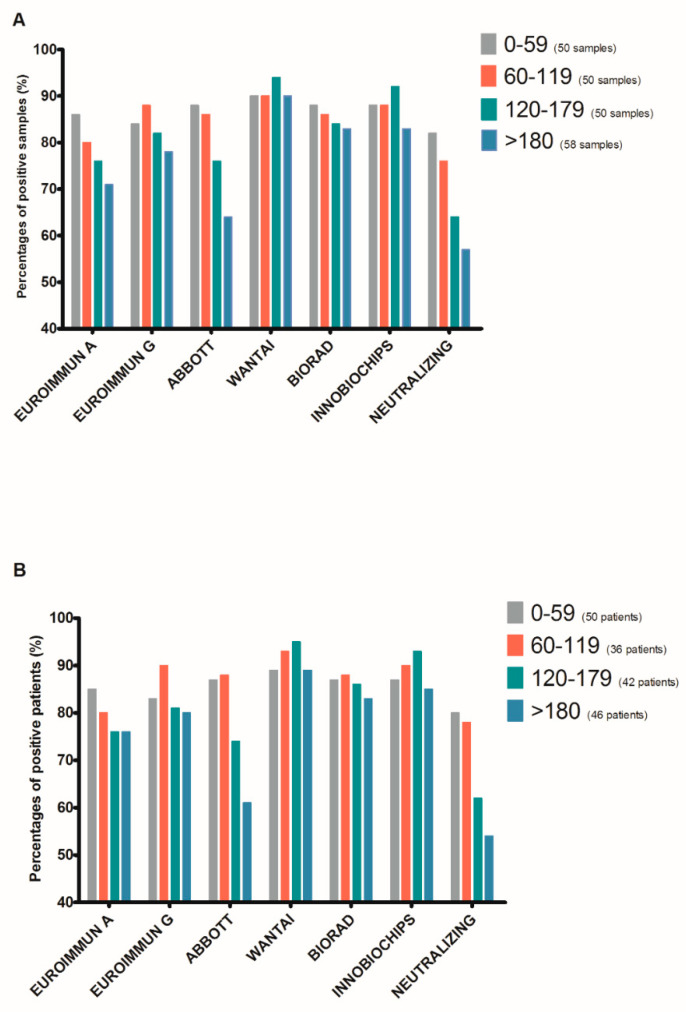
Percentage of SARS-CoV-2 seropositive specimens (**A**) or patients (**B**) obtained with various serological assays.

**Figure 2 jcm-10-01815-f002:**
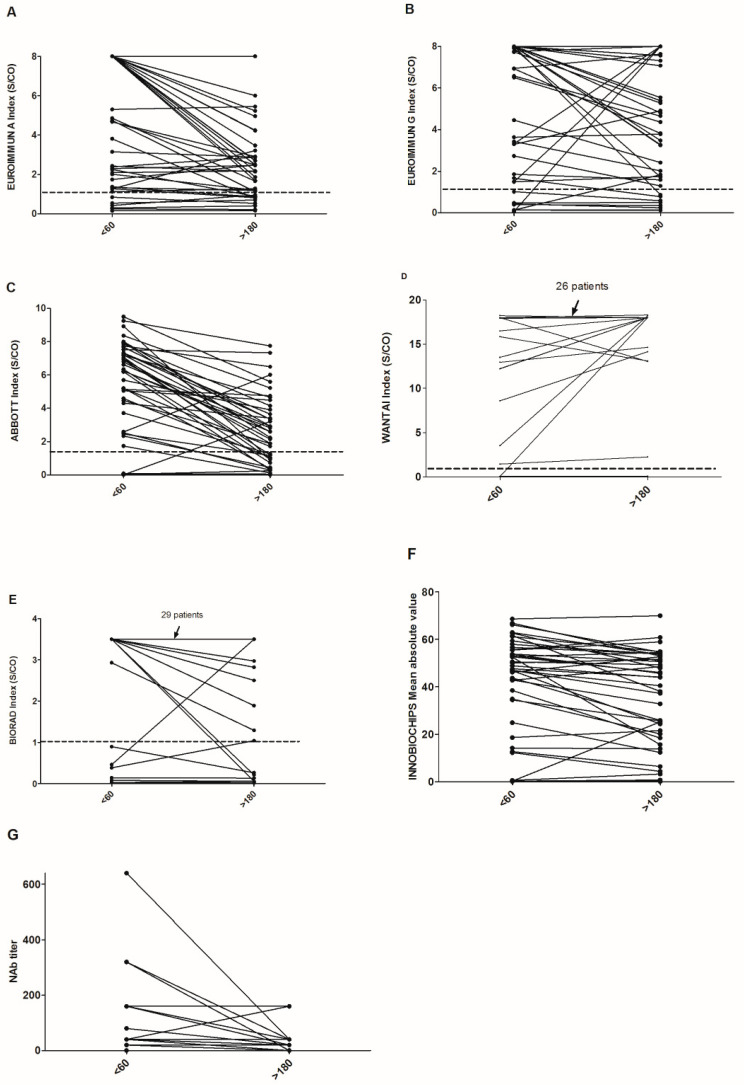
Kinetics of the raw results of the various assays (**A**–**G**) by comparing the values before Day 60 and after Day 180 post-PCR. The positivity threshold for each assay is indicated by a dotted line.

**Figure 3 jcm-10-01815-f003:**
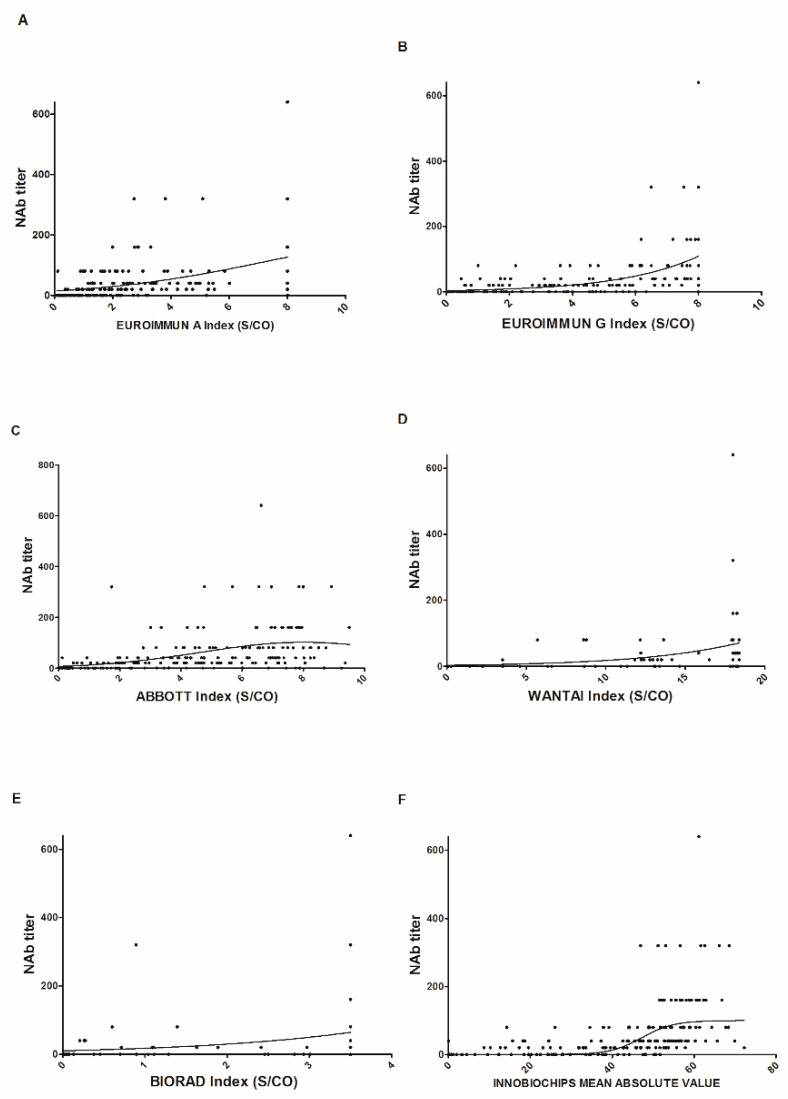
Relationship between the index values found for each assay and the neutralizing titers for the 208 samples (**A**–**F**). The solid lines represent the nonlinear fit of the neutralizing titles according to the indexes found for each assay.

**Table 1 jcm-10-01815-t001:** Cohort characteristics.

Number of patients	61
Female	36
Male	25
Age (Years):	
Median	74
Range	26–98
>65 years	41
Hospitalized patients	27
Nonhospitalized patients	34
Immunocompromised patients	6 (2 kidney transplant, 2 bone marrow transplant, 2 chemotherapy
Numbers of samples (days post-PCR):	
0–59	50
60–119	50
120–179	50
≥180	58
	(range: 180–237)
Numbers of patients (days post-PCR):	
0–59	50
60–119	36
120–179	42
≥180	46

**Table 2 jcm-10-01815-t002:** Percentage of reversion at Day 180 post-PCR.

Serological Assay	All	Female	Male	*p*-Value	Inpatients	Outpatients	*p* Value	>65	≤65	*p*-Value
**EUROIMMUN A**	24	27	20	0.58	9	37	0.02	23	25	0.9
**EUROIMMUN G**	20	19	20	0.94	9	29	0.09	13	31	0.14
**ABBOTT**	39	42	35	0.61	36	42	0.71	27	62	0.02
**WANTAI**	11	15	10	0.26	5	17	0.19	7	19	0.21
**BIORAD**	17	23	10	0.24	14	21	0.52	10	31	0.07
**INNOBIOCHIPS**	15	19	10	0.74	5	25	0.05	10	25	0.18
**Neutralizing antibodies (titer = 0)**	46	50	40	0.49	41	50	0.53	47	44	0.85

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
