# Peer review of "Longitudinal Analysis and Comparison of Six Serological Assays up to Eight Months Post-COVID-19 Diagnosis"

_jcm, 2021, doi:10.3390/jcm10091815_

Round 1
Reviewer 1 Report
Congratulations, this is a very interesting manuscript, thank you for the opportunity to review it!
In Table 1 you mention 10 immunocompromised patients included in the study; please provide further information about the antibody titers of these patients compared to other patients.
There is a typo in Table 1: bone narrow transplant -> bone marrow transplant.
Please make the appropriate additions and the correction.
Reviewer 2 Report
The manuscript reports the comparison of 6-serological assays for the evaluation of COVID-19 diagnosis up to 8-months. There are some points that need to be clarified and better described, especially as regards the approaches adopted in analysing the data and in the corresponding results interpretation, as pointed out below
- Authors declared a sample size of 61 patients; however, from Table 1 it seems that Authors worked on 50 samples for each time points: please better clarify this issue also in respect to Figure 1.
- the subset of patients for which all the 4 time points were available is an important information that should be reported
- are there more than one sample for each time for each patient?
- Table 1:
- which is the range of the age variable? Please comment about the choice of the 65 yrs as cut-off value to dichotomize the age
- Authors should reports in addition to the number of samples the corresponding number of patients from which the sample arise
- Minor: please clarify the meaning of “inpatients” and “Outpatients”
- Authors should justify the ranges (~50-60 days) adopted to categorize the time post-PCR? Which is the range of the “>180 days” category? What about the results by considering the time post-PCR on their continuous scale?
- It is not clear which statistical test Authors adopted in evaluating statistical significance of the 120-170 days period and >180 days (section 3.1). In addition, a p-value of 0.05 is not statistically significant according to that declared in the data analysis section.
- Section 2.4 should be improved with more details and pertinent reference. Has Authors take into account any correction for multiple comparisons? It is not clear if Authors also performed a longitudinal analyses and which method they adopted.
- As it seems that multiple measurements are available for each patients, how Authors manage data correlation within patient in the longitudinal analysis?
- Authors should reports in the text or figures the p-values of the association depicted in Suppl. Figure 1. A clear figure legend should also be inserted.
- Figure 2: add a figure legend. What about the results from the analysis of the data on this continuous scale?
- Figure 2: Authors should justify why they compared only samples arising from the first and last time points (<60 days vs >180 days) and not all the longitudinal profile. In any case a p-value corroborating the report results could aid the data interpretation.
- Table 2: p-values of non-significant results should be reported instead of “NS” together with the detail of the test adopted to analyse the data.
- Figure 3: add a figure legend. What isthe continuous line inside the graphs?
- Conclusions: According to which data Authors declared that “The current serological tests show equivalent sensitivity”? How they assessed the assay’ sensitivity?
Minor:
- The positivity’s thresholds should be reported in main text
Reviewer 3 Report
The manuscript entitled “Longitudinal analysis and comparison of six serological assays up to eight months post-COVID19 diagnosis” by Aubry et al describes the seropositivity in individuals tested positive for COVID-19 using six different assays. The study was performed up to 8 months after the detection of positive RT-PCR test for SARS-CoV-2. Overall, the manuscript is interesting and covers a very important topic. However, English language has to be improved throughout the manuscript.
Data include mostly higher age group. The results can also vary with age. Did authors evaluate the results in young individuals? In page 4, authors have discussed a little bit about the people with age <65. However, this is pretty vague analysis. <65 might be even 64, so it is hard to conclude anything here. Authors could take the range 20-40, 40-60 yr.
Authors have made a comparison between different assays. What control did authors use for all these assays as well as neutralization assay? If a commercial SARS-CoV-2 antibody is used for all these assays, is there still discrepancy?
Neutralizing antibody titers are going down as well by ~50% after almost 6 months. On the basis of this results, what do authors think about the antibody titers after vaccination?
How did authors verify their Neutralization assay? Authors should have verified the pseudotyped particles by immunoblotting, live cells imaging or any other method? The results should be included.
Round 2
Reviewer 2 Report
I have read the revised version of the manuscript and the Authors’ response. Authors have responded to the comments that I have raised. There are still few points that need to be better clarified as reported below.
1. From Table 1 is now clear that for some patients are available multiple samples at each considered time-points. I suggest to report in the main text which sample was considered in the analysis (and Figure 1) in these cases, as reported by the Authors in the Review’ reply.
2. Instead of reporting “p<0.05” it would be better to report the point estimated p-value as reported by the Authors in the Review’ reply
3. Statistical analysis should clearly report which are the variable considered in the Chi Square/Fisher analysis (which are the qualitative variables?); coherently in the results section the detail of the test should be places together with the p-value (i.e. Fisher p-value: 0.05).
4. Figure 1 in a color-style could aid in the visualization of the data
5. In Figure 2 are descriptively reported the trends of the titers arising from the “<60” vs “>180” days. Authors state that “There was a clear decrease of the indexes and titers, with considerable heterogeneity between assays”: which test they adopted to evaluate this trend? The number of patients included in this Figure should be also reported.
